# The Challenge of Evaluating Response to Peptide Receptor Radionuclide Therapy in Gastroenteropancreatic Neuroendocrine Tumors: The Present and the Future

**DOI:** 10.3390/diagnostics10121083

**Published:** 2020-12-12

**Authors:** Virginia Liberini, Martin W. Huellner, Serena Grimaldi, Monica Finessi, Philippe Thuillier, Alfredo Muni, Riccardo E. Pellerito, Mauro G. Papotti, Alessandro Piovesan, Emanuela Arvat, Désirée Deandreis

**Affiliations:** 1Nuclear Medicine Unit, Department of Medical Sciences, University of Turin, 10126 Turin, Italy; grimaldi@cittadellasalute.to.it (S.G.); mfinessi@cittadellasalute.to.it (M.F.); philippe.thuillier@chu-brest.fr (P.T.); desiree.deandreis@unito.it (D.D.); 2Department of Nuclear Medicine, University Hospital Zurich, University of Zurich, 8091 Zurich, Switzerland; martin.huellner@usz.ch; 3Department of Endocrinology, University Hospital of Brest, 29200 Brest, France; 4Department of Nuclear Medicine, S.S. Biagio e Antonio e C. Arrigo Hospital, 15121 Alessandria, Italy; amuni@ospedale.al.it; 5Department of Nuclear Medicine, AO Ordine Mauriziano di Torino, 10128 Turin, Italy; repellerito@mauriziano.it; 6Pathology Unit, City of Health and Science University Hospital, 10126 Turin, Italy; mauro.papotti@unito.it; 7Department of Oncology, University of Turin at Molinette Hospital, 10126 Turin, Italy; 8Department of Endocrinology, A. O. U. Città della Salute della Scienza of Turin, 10126 Turin, Italy; apiovesan@cittadellasalute.to.it; 9Oncological Endocrinology, Department of Medical Sciences, University of Turin, 10126 Turin, Italy; emanuela.arvat@unito.it

**Keywords:** peptide receptor radionuclide therapy, PRRT, ^68^Ga-labeled somatostatin analogue, ^18^F-FDG, response assessment, RECIST, SWOG, WHO, neuroendocrine tumors, NET

## Abstract

The NETTER-1 study has proven peptide receptor radionuclide therapy (PRRT) to be one of the most effective therapeutic options for metastatic neuroendocrine tumors (NETs), improving progression-free survival and overall survival. However, PRRT response assessment is challenging and no consensus on methods and timing has yet been reached among experts in the field. This issue is owed to the suboptimal sensitivity and specificity of clinical biomarkers, limitations of morphological response criteria in slowly growing tumors and necrotic changes after therapy, a lack of standardized parameters and timing of functional imaging and the heterogeneity of PRRT protocols in the literature. The aim of this article is to review the most relevant current approaches for PRRT efficacy prediction and response assessment criteria in order to provide an overview of suitable tools for safe and efficacious PRRT.

## 1. Introduction

Neuroendocrine neoplasms (NENs) are a heterogeneous group of malignancies represented by different histological subtypes, primary locations and functional status [1]. NENs range from well-differentiated neuroendocrine tumors (NETs), which are mainly indolent neoplasms, to poorly differentiated neuroendocrine carcinomas (NECs), which are highly aggressive cancers with poor prognosis [2]. NENs originated in the gastroenteropancreatic (GEP) tract are graded according to Ki-67 index and mitotic count, which represents features of the proliferative activity of the tumor (Table 1).

The mutational status also has an important impact on NEN behavior. Several studies have demonstrated a strong correlation between well-differentiated NETs and MEN1, DAXX and ATRX mutations, whereas NECs usually carry TP53 or RB1 mutations [3,4,5]. According to the National Cancer Institute’s Surveillance, Epidemiology and Results (SEER) Program, the incidence of NENs increased 6.4-fold from 1973 (1.09 per 100,000) to 2012 (6.98 per 100,000) [6], owing to an increasing awareness of NEN occurrence and also due to a more effective identification of these tumors. A remarkable feature of NENs is the expression of somatostatin receptors (SSTRs) in well differentiated tumors, with SSTRs type 1 and type 2 being present in the vast majority of GEP-NENs, while SSTRs type 3 and type 5 are expressed by approximately 60% of cases, and SSTR type 4 only rarely [7]. The knowledge of histopathological and molecular characteristics of NENs as well as the availability of more accurate diagnostic tools and therapeutic options, allows for a personalized approach to these diseases, with potential benefits in treatment response and survival.

In vivo imaging of SSTR expression in NENs has become feasible since the development of [^123^I]I-labelled tyr-3-octreotide in 1989 [8,9], when Krenning et al. documented for the first time positive [^123^I]I-labelled tyr-3-octreotide scans obtained for two meningiomas, two gastrinomas and one carcinoid [9]. In the last decade, the accuracy in NEN detection by [^111^In]In-pentetreotide (Octreoscan^®^) single-photon emission computed tomography (SPECT)/computed tomography (CT) has been surpassed by [^68^Ga]Ga-DOTA-labelled somatostatin analogue positron emission tomography (PET)/CT. Compared to SPECT radiopharmaceuticals, somatostatin analogue PET/CT also has the advantage of lower radiation exposure, earlier and shorter acquisition times, higher spatial resolution and the possibility of tracer uptake quantitation [10,11]. The main [^68^Ga]Ga-DOTA-labelled somatostatin analogues clinically available today are [^68^Ga]Ga-DOTA-TATE (DOTA, Tyr(3)-octreotate), [^68^Ga]Ga-DOTA-NOC (DOTA,1-Nal(3)-octreotide) and [^68^Ga]Ga-DOTA-TOC (DOTA, D-Phe1, Tyr (3)-octreotide).

Peptide receptor radionuclide therapy (PRRT) has been proven to be an effective systemic treatment in the clinical management of patients with advanced, metastatic or inoperable, slowly progressing NENs with high somatostatin receptor expression. The principle behind PRRT efficacy is the dual component of the radiopeptide: (1) the somatostatin receptor ligand that binds the specific receptor (SSTR1-5, especially SSTR2) overexpressed on the surface of neuroendocrine tumor cells, allowing its internalization into the tumor cells and (2) the high energy of the radioactive β-particle (^90^Y or ^177^Lu) labeled to a somatostatin receptor (SSTR) ligand, yielding cell apoptosis through direct or indirect DNA damage of target cells (self-dose) or neighboring cells (cross-fire effect) [12]. In the last few years, DOTATATE (DOTA, Tyr(3)-octreotate), labeled either with [^117^Lu]Lu or [^90^Y]Y as radionuclides, was the peptide most widely used, owing to its higher SSTR2 affinity compared to DOTATOC (DOTA, D-Phe1, Tyr (3)-octreotide) and DOTANOC (DOTA, 1-Nal(3)-octreotide), which is present in the vast majority of GEP-NENs. These two radiopeptides share similar radiobiology and pharmacokinetic aspects, such as fast blood clearance and urinary elimination, low whole-body radiation exposure and a high absorbed dose to the spleen, the kidney and the liver, but they have different toxicity profiles. An important advantage of [^117^Lu]Lu is its partial decay into γ photons (E = 113 KeV and 208 KeV), allowing the acquisition of SPECT/CT images, which is useful for dosimetry and immediate therapy response assessment [13,14,15].

As of today, [^117^Lu]Lu-DOTA-TATE has almost completely replaced [^90^Y]Y-DOTA-TATE and [^90^Y]Y-DOTA-TOC. In 2017, the phase III NETTER-1 trial showed that [^117^Lu]Lu-DOTA-TATE treatment resulted in longer progression-free survival and a higher response rate compared to high dose somatostatin analogues in patients with advanced midgut neuroendocrine tumors (NETs) and disease progression after first-line somatostatin analogue therapy [16]. Owing to this result, [^117^Lu]Lu-DOTA-TATE (Lutathera^®^) was approved by the US Food and Drug Administration (FDA) and the European Medicine Agency (EMA) in the year 2018 for the treatment of inoperable or metastatic well-differentiated gastroenteropancreatic) NETs with disease progression. More recently, α particle (i.e., actinium-225 (225Ac) or bismuth-213 (213Bi) labeled somatostatin receptor (SSTR) ligand through a chelator (dodecane tetra-acetic acid (DOTA)) have been applied in clinical trials [17]. An overview of characteristics of theragnostic radionuclides in patients with NENs is given in Table 2.

In order to achieve a clinical benefit, both appropriate patient selection and accurate response assessment to PPRT are essential. In fact, disease progression during therapy is reported in approximately 20–30% of patients, and in approximately 10% of them within 6 months to 1 year after PPRT [16,21]. Furthermore, PRRT treatment is not exempt from toxicity. The critical organs are the bone marrow and the kidneys. More specifically, the kidneys are the dose-limiting organs. Therefore, an intravenous administration of an amino acid solution consisting of lysine and arginine, prior to PRRT, can partially protect the kidneys from radiation damage [17]. In 2018, Baum et al. [22] presented the outcome of personalized [^117^Lu]Lu and/or [^90^Y]Y PRRT in a cohort of 1048 patients with NENs. According to the Common Terminology Criteria for Adverse Events (CTACAE criteria), grade 3 and 4 adverse events were recorded in <1% of patients. However, myelodysplastic syndrome (MDS) or leukemia and chronic kidney disease developed in 22 (2%) and 5 (0.4%) patients, respectively. In a cohort of 807 patients with NENs, Bodei et al. [23] reported grade 3 and 4 hematological toxicity, myelodysplastic syndrome and grade 4 renal failure in approximately 10%, 2–4% and <1% of patients, respectively.

The identification of biomarkers to assess PRRT efficacy and avoid patient toxicity is crucial, but also challenging in a heterogenous group of tumors such as NENs. The current parameters (biomarkers, such as chromogranin A and RECIST 1.1) used for PRRT response assessment are considered suboptimal based upon the Delphic consensus assessment for GEP-NENs [24], owing to the variability in somatostatin receptor expression, histology and the characteristic slow growth of these tumors. Functional imaging in particular and new approaches in image analysis could play a key role as prognostic biomarkers and for the therapy response assessment. Hence, standardization is needed.

The aim of this article is to review biomarkers for PRRT efficacy prediction and response assessment criteria proposed in the literature, with a particular focus on the role of clinical imaging (both anatomical and functional).

## 2. Materials and Methods

A literature search was performed in the time period between January 2010 and September 2020, including PubMed, Scopus, Embase and Google Scholar databases, in accordance to the Preferred Reporting Items for Systematic Review and Metanalysis (PRISMA) guidelines [25]. The following search terms were applied: (“PRRT” OR “peptide receptor radionuclide therapy”) AND (“177Lu” OR “90Y”) AND (“NET” OR “neuroendocrine tumor*”) AND (“therapy response” OR “assessment” OR “response” OR “efficacy”) AND (“RECIST1.1” OR “PERCIST” OR “positron emission tomography” OR “computed tomography” OR “magnetic resonance” OR “PET” OR “CT” OR “MR” OR “DOTATOC” OR “DOTANOC” OR “DOTATATE” OR “FDG” OR “DOPA” OR “biomarker” OR “liquid biopsy” OR “NETest” OR “PPQ” OR “dosimetry*”).

Additional filters, such as English language, original article and/or research article, and studies including only humans and with 10 or more subjects, were used. Reviews, clinical reports, meeting abstracts and editorial comments were excluded. Due to the comparably high number of studies available on this topic, an additional exclusion was based on citation threshold >20 citations for years 2010–2018 and >10 citation for years 2019–2020. The authors reviewed abstracts and read the full text of all the included articles. Relevant studies that were not obtained by the original search were included through cross-references. A graphical representation of the search and review strategy is presented in Figure 1.

### 2.1. Definition of a Biomarker

The National Institutes of Health Biomarkers Definitions Working Group and the International Programme on Chemical Safety has defined a biomarker as a substance, structure or process that is objectively measured in the body (or its products) and is evaluated as an indicator of normal biological processes, pathogenic processes or pharmacologic responses to a therapeutic intervention, influencing or predicting the incidence of outcome or disease [26,27,28]. The National Institute of Health Biomarkers Definitions Working Group has also described biomarkers as: (1) diagnostic tools to identify patients with disease or abnormal conditions; (2) staging tools to classify the extent of disease; (3) prognostics of patient outcome for a specific disease; (4) predictive of clinical response to a specific intervention [28].

Prognostic biomarkers are generally derived from observational data, while predictive biomarkers are usually derived from clinical trials comparing a group of patients treated with a control group of patients and biomarker expression. However prognostic and predictive biomarkers are frequently confused, even if they can occasionally be both at the same time [29]. In neuroendocrine tumors, one can divide the most widely used biomarkers into clinical biomarkers, pathological tissue biomarkers, circulating biomarkers, genomic multianalyte biomarkers and imaging biomarkers (including dosimetry).

The response to treatment should be based on a careful evaluation of all these biomarkers. In particular the objective tumor response (OTR), the patient’s health status and the biochemical response in terms of reduction/increase in tumor markers should be used for judging treatment outcome [30]. The OTR can be assessed through the analysis of the variation of measurable disease by imaging biomarkers between baseline and another timepoint during or after treatment.

### 2.2. Clinical Biomarkers: Quality of Life and Symptom Control

In recent years, the patient’s health status (PHF) has become more relevant for the evaluation of therapy outcome. This is particularly the case in slow-growing tumors, such as NENs, where the assessment of the OTR could be difficult and the impact of therapy is also related to an improvement in clinical symptoms by inhibiting hormonal hypersecretion. The most common scales used for this purpose are the Karnofsky Performance Status (KPS) scale and the health-related quality of life (HRQoL) scale [31]. In 2008, a retrospective study by Kwekkeboom et al. [32] on the efficacy and toxicity of PRRT treatment in 500 patients with GEP-NETs reported for the first time a KPS greater than 70 as a significant predictor for favorable treatment outcome after PRRT (*p*-value 0.05).

More recently, HRQoL was introduced as a prognostic parameter for PRRT response. Patients enrolled in the NETTER-1 trial were evaluated by the European Organization for Research and Treatment of Cancer (EORTC) quality-of-life questionnaires (QLQ) C-30 and GEP-NETs (evaluating gastrointestinal symptoms, any factors related to cancer, psychosocial problems, treatment side effects and other events) at baseline and every 12 weeks until disease progression. A clinically and statistically significant improvement in HRQoL and a significant reduction of symptoms, such as diarrhea, pain and fatigue, were observed in patients submitted to PRRT compared to those treated with high-dose octreotide [33].

Other groups have recently studied the QoL impact in clinical routine cohorts. In a retrospective study of 34 patients with metastatic and functional pancreatic NET (pNET), Zandee et al. reported that symptoms were well controlled in 78% of patients (18/23) with progressive disease, defined as a reduction of symptoms and/or circulating biomarker in patients with a functioning pNET (insulinoma, gastrinoma, VIPoma and glucagonoma). Additionally, 71% of patients with uncontrolled symptoms showed a reduction of symptoms after [^177^Lu]Lu-PRRT. After PRRT, the median progression-free survival after the first treatment was 18.1 months for the entire cohort. Follow-up with EORTC QLQ-C30 was available for 22 patients. All of these 22 patients reported a substantial increase in QOL at 3 months after the final PRRT cycle, with a concomitant increase in physical, emotional and social functioning [34].

Marinova et al. [35] have evaluated the QoL (EORTC QLQ-C30) in a retrospective cohort of 68 patients with advanced pancreatic NET treated with PRRT. Compared to baseline, QoL was significantly improved at the end of the study, particularly due to increased global health status (*p* = 0.008) and improved social functioning (*p* = 0.049). Furthermore, several symptoms (such as fatigue, nausea and vomiting, dyspnea, appetite loss and constipation) were significantly alleviated even after the first cycle of PRRT. QoL can thus be considered a useful tool for monitoring therapy benefits and treatment side effects.

Martini et al. [36] evaluated HRQoL in 61 patients with GEP-NETs from the first peptide receptor radionuclide therapy (PRRT) to the first restaging, and compared the scores with general population (GP) norms. At baseline, patients with GEP-NETs presented the same symptoms as those reported in the study of Marinova et al. (diarrhea, fatigue, appetite loss, reduced global HRQoL), resulting in a lower HRQoL score compared to the HRQoL score reported for the general population (GP). Contrary to the results of the previous study, except for diarrhea and appetite loss, patient scores at the first restaging did not reach GP levels.

### 2.3. Pathological Tissue Biomarkers

#### 2.3.1. Ki67 Expression and Grading

Ki-67 expression is considered a prognostic biomarker and is related to the tumor grade of NETs. However, Ki67 immunohistochemical count depends on a large number of variables, such as type of biopsy, cold ischemia time, type of fixative, fixation time, Ki67 count based on visual scoring or by automated digital analysis of the same field of view [3,37]. Moreover, tumors are typically characterized based on a single biopsy specimen, which bears the inherent risk of an underestimation of the tumor grade [38]. It is known that tumor heterogeneity is subject to space (intertumoral and intratumoral heterogeneity) and time (more aggressive cell clones developing over time) [39,40]. Thus, grading heterogeneity between primary and secondary lesions is not negligible. Performing multiple-lesion biopsies may help to account for this heterogeneity. However, biopsy is an invasive procedure, and may not be feasible for every lesion [41]. Finally, the way Ki-67 index is assessed (random counting versus determination in the area of highest labeling) is an additional potential source of discrepancies. For all these reasons, the prognostication of PRRT response by Ki-67 expression is discussed controversially in the literature.

In a retrospective study on a cohort of 68 patients with pNETs treated with [^117^Lu]Lu-octreotate, Ezziddin et al. [42] reported that G1 was associated with longer PFS (*p* = 0.04) and OS (*p* = 0.044) compared to G2 tumors (45 versus 28 months for PFS, respectively). However, the larger prospective cohort of the NETTER-1 study [16] showed no difference between G1 and G2 proliferation status with regard to PRRT response; patients treated with [^117^Lu]Lu-DOTA-TATE resulted in markedly longer PFS compared with patients treated with high-dose octreotide LAR, regardless of G1 and G2 proliferation status. On the other hand, in retrospective studies focusing on patients with G3 tumors, the median PFS was much lower compared to patients with G1 and G2 NETs in other studies [43,44]. Stratifying the PFS according to the grading, Sorbye et al. [45] recently reported a PFS of 19, 11 and 4 months for G3 NET, low (<55%) Ki-67 NEC and high (>55%) Ki-67 NEC, respectively.

#### 2.3.2. Primary Origin

In a large registry study, Yao et al. [46] found that the median overall survival among patients with metastatic NETs varies from 56 months (jejunal/ileal tumors) to 5 months (colon tumors). With regard to PRRT outcome, compared to Ki-67 expression, the impact of the site of neuroendocrine tumor origin on PRRT outcome seems clearer, as shown by Brabander et al. [47] For the entire groups of 443 NET patients, the median OS was 63 months and the median PFS was 29 months. Patients with a primary pancreatic NET had the longest OS (71 months), followed by midgut, unknown and bronchial NET with an OS of 60, 53 and 52 months, respectively. Moreover, the OS was significantly shorter in patients with liver or bone metastases at baseline, underlining that also the stage and liver involvement may have an impact on PRRT efficacy.

### 2.4. Circulating Biomarkers

#### 2.4.1. Serum Biomarkers

Neuroendocrine tumors are characterized by the production of several circulating biomarkers, that can be divided into:general circulating biomarkers, such as chromogranin A (CgA), and neuron-specific enolase (NSE);specific circulating biomarkers, such as serotonin and its metabolite 5-hydroxyindole acetic acid (5-HIAA), insulin, glucagon, vasoactive intestinal peptide and gastrin [1,48], inducing specific clinical symptoms.

Among these circulating biomarkers, chromogranin A (CgA) is the most commonly used. CgA is an acidic glycoprotein of 439 amino acids and has a molecular mass of 48 kDa. It is released from NET cells into the blood by exocytosis. Serum CgA can be increased in several non-tumoral conditions (i.e., gastritis, during proton pump inhibitor therapy, systemic hypertension, renal and liver insufficiency, pancreatitis) and also in presence of non-neuroendocrine tumors, such as prostate cancer and differentiated thyroid cancer. Moreover, some NETs (i.e., insulinoma, MEN-1 associated tumors, poorly differentiated NETs) have a low CgA expression [49,50,51,52]. Despite this, CgA can be clinically useful. In a recent meta-analysis on the role of CgA as a diagnostic marker of NETs, Yang et al. reported an overall sensitivity and specificity of 73% and 95%, respectively, with an area under the receiver-operating characteristic (ROC) curve of 0.90 [53]. Several studies also highlighted the predictive role of CgA in monitoring disease progression and therapy response assessment. In 2015, Sabet et al. [54] evaluated the efficacy of PRRT in 61 consecutive patients with small intestinal NETs with both objective tumor response (OTR) and with the patient’s health status (PHF) parameters. They reported that an increased plasma level of chromogranin A (CgA > 600 ng/mL) was associated with earlier tumor progression and that patients with carcinoid symptoms (due to serotonin hypersecretion) had shorter progression-free survival (PFS) after PRRT.

This suggests that CgA may be considered a surrogate marker for tumor burden, even if its prognostic role on PRRT response assessment has yet not been evaluated prospectively. To our knowledge, only the RADIANT-2 trial on everolimus with octreotide LAR [55] has reported a reduction in serum chromogranin A and 5-hydroxyindoleacetic acid levels to be associated with improved PFS.

Another general circulating biomarker with a potential predictive role for PRRT response is the neuron-specific enolase (NSE). NSE is a glycolytic enzyme typically present in the cytoplasm of neurons and neuroendocrine cells. Literature data are still limited and mostly focus on everolimus response assessment [55,56,57]. Today, only one study by Ezziddin et al. on a consecutive cohort of 74 patients with metastatic GEP-NET after PRRT [58] reported a predictive role of NSE. The authors stated that a baseline plasma level of NSE > 15 ng/mL independently predicted shorter overall survival (hazard ratio, 2.2, *p* = 0.035) on multivariate analysis, as well as a Karnofsky performance score <70% (hazard ratio, 3.1, *p* = 0.007), a hepatic tumor burden >25% (hazard ratio, 2.9, *p* = 0.017) and G2 (hazard ratio, 2.8, *p* = 0.044).

Finally, two different groups [59,60] recently proposed a new general circulating biomarker in order to predict the PRRT outcome, the inflammation-based index (IBI), which is derived from serum C-reactive protein (CRP), and albumin levels. Patients with normal CRP (<10 mg/L) and albumin (>35 g/L) levels are assigned a score of 0 (IBI 0), patients with one abnormal parameter are assigned a score of 1 (IBI 1) and patients with both elevated CRP and hypoalbuminemia are assigned a score of 2 (IBI 2). Pauwels et al. [60] showed that normal baseline IBI and high [^68^Ga]Ga-DOTA-TOC tumor uptake predict a better outcome in NET patients treated with [^90^Y]Y-DOTA-TOC.

Inappropriately elevated levels of specific circulating biomarkers (such as 5-HIAA, insulin, glucagon, vasoactive intestinal peptide and gastrin) are responsible for specific clinical symptoms, and their measurement on blood and/or urine samples leads to the diagnosis of secreting or functional NETs, which must be either treated or monitored carefully. However, even if a post-therapy reduction of such specific biomarkers might reflect the reduction of the tumor burden owing to effective treatment (and for this reason their assessment is mandatory in functional NETs), there is no strong evidence on their role in PRRT response assessment. This is mainly due to the fact that tumors with secretory products represent less than 10% of all NETs and hence the available data in this field is still limited [61]. Nevertheless, all patients should be screened for specific biomarkers (such as 5-HIAA, insulin, glucagon, vasoactive intestinal peptide and gastrin), particularly if specific clinical symptoms are suspected.

The only specific circulating biomarker with a possible predictive role for PRRT response is 5-hydroxyindolaceric acid (5-HIAA), which represents the main metabolite of the amine derivate serotonin secreted by the enterochromaffin cells located in the small intestine. Serotonin can be overproduced by small intestinal NETs, but the typical carcinoid syndrome occurs only in approximately 18% of patients with small intestinal enterochromaffin tumors. Moreover, in most cases, patients with liver metastases present a carcinoid syndrome as a direct consequence of the inability of the liver to metabolize the overexpressed serotonin [1]. Sabet et al. [54] reported that patients suffering from carcinoid symptoms related to serotonin hypersecretion have shorter PFS after PRRT. Nevertheless, these data still need to be confirmed.

#### 2.4.2. Genomic Multianalyte Biomarkers

In recent years, liquid biopsy has evolved as a promising biomarker technique. Liquid biopsy refers to the analysis of both circulating tumor cells (CTC) and circulating tumor DNA (ctDNA) detected through PCR techniques. Liquid biopsy may facilitate tumor understanding and management by an earlier detection and improved characterization of a tumor beyond the aforementioned limitations of classic biopsy. It may help establish a tailored therapy approach, and may accurately predict response and toxicity related to therapy [62,63].

Since 2013, liquid biopsy has been explored also in the NETs field, with the introduction of the NETest, a multi-transcript molecular signature of 51 specific NET genes for PCR-based blood analysis [64]. This tool is able to differentiate NENs from controls with high PPV and NPV (>90%) and could confirm GEP-NENs origin if CgA levels were low in a cohort of 130 blood samples (NENs: *n* = 63) and two independent validation sets (Set 1 [*n* = 115, NENs: *n* = 72]; Set 2 [*n* = 120, NENs: *n* = 58]).

In another prospective study by Modlin et al. [65], using clinical blood samples [n=159] obtained from 111 patients with stable disease (SD) and 48 patients with progressive disease (PD)] [65], the original score (0 to 8) derived from the blood-based gene test was weighted with the additional information about disease status (SD or PD), which led to the development of a new 0–100% activity score. In this study, SD and PD were assessed based on standard clinical criteria and on changes (RECIST criteria [66]) in radiological and nuclear medicine images: PD was defined as any increase in tumor burden (any individual lesion) and stable disease (SD) was defined as no increase in tumor burden, including both unchanged or responding lesions. The new score derived thereby predicts disease status as stable (SD) or progressive (PD) with high accuracy (87.4%), identifying PD in patients with a score >45%.

These results have been further validated in several independent studies [67,68,69,70,71,72]. In particular, van Treijen et al. [71] emphasize the potential role of liquid biopsy as a marker of persistent disease after therapy or relapse during follow-up. Liu et al. [69] showed a significant correlation of the NETest score with disease status determined by RECIST in 100 patients with NETs (68% GEP, 20% lung and 12% of unknown origin), after a 6–12 month follow up. Based on their work, they slightly adjusted the cut-off for progressive disease (>40%), stratifying patients into three risk groups for tumor activity according to the NETest score (low ≤ 40%; moderate/intermediate 41–79%; and high ≥ 80%). A recent meta-analysis on NETest [73] identified a diagnostic accuracy for NET disease of 95–96%; an accuracy of 84.5–85.5% in differentiating stable disease from progressive disease, an accuracy of 91.5–97.8% as a marker of disease trend (taking as reference the biomarkers used in normal practice) and an accuracy of 93.7–97.4% as an interventional/response biomarker.

More recently, Bodei et al. [74] developed an algorithm that integrates blood-derived NET-specific gene transcripts with tissue Ki-67 values (AUC: 0.90±0.07, irrespective of tumor origin) to predict PRRT responders versus non-responders; their results showed that the algorithm was able to predict PRRT response with an high accuracy (94%). Their study was conducted in 78 patients with GEP and bronchopulmonary (BP) NET. This algorithm was subsequently defined as the PRRT predictive quotient (PPQ) by Bodei et al. [75] and validated as a highly specific predictor of PRRT efficacy, yielding an accuracy of 95% in three independent [^177^Lu]Lu-PRRT-treated cohorts from IRST Meldola (Italy), Zentralklinik Bad Berka (Germany) and Erasmus Medical Center, Rotterdam (The Netherlands). However, the PPQ has yet not been validated as a prognostic marker of survival.

In 2020, Bodei et al. [76] prospectively evaluated the usefulness of both NETest and PPQ in a cohort of 158 patients with GEP or lung NETs enrolled for [^177^Lu]Lu-PRRT-treatment. Patients were divided into responders and non-responders based on treatment response assessment by RECIST 1.1 criteria [77]. NETest was performed before PRRT, after each cycle and during follow-up. NETest score significantly (*p* < 0.0001) decreased in responders, while it remained high in non-responder patients (*p* < 0.0005). Moreover, during follow-up, NETest scores (>40 for progressive and <40 for stable disease) significantly correlated with median PFS. Response prediction by PPQ was accurate in 97% of cases, and NETest score significantly (*p* < 0.001) decreased in PPQ-predicted responders and remained high in PPQ-predicted non-responders. Moreover, CgA serum levels did not reflect response to PRRT. This study further validates both the use of the PPQ to predict response to PRRT and the use of the NETest to monitor PRRT response as surrogate biomarkers of tumor burdens routinely assessed trough radiological and nuclear medicine images. These tests applied in clinical practice are expected to individualize treatment strategies, identify patients who may benefit from PRRT and patients in whom PRRT is not effective and who need different management.

Although these results are promising, it must be pointed out that liquid biopsy in general is not exempt from limitations: liquid biopsy is not yet considered a standard testing procedure and further prospective trials will be necessary for this purpose; there is still a lack of evidence on the clinical utility of these tests, considering that liquid biopsy cannot replace tissue biopsy and/or conventional or functional imaging for staging and restaging; test sensitivity and accuracy is strictly correlated to additional mathematical analysis by use of a machine learning method and deep neural network analysis, and this must be outlined in the evaluation of these tests and a higher level of standardization is desired to ensure a more robust test result. Finally, these tests are expensive, are typically not refunded by health insurance and their availability is limited.

### 2.5. Imaging Biomarkers

Besides clinical and molecular biomarkers, both radiological (CT and MR) and functional (PET/CT, PET/MR and SPECT/CT) imaging plays a fundamental role in the management of patients with NETs in a PRRT setting. This holds true both for inclusion criteria (inoperable and progressive disease and confirmed overexpression of somatostatin receptors on target lesions) and the evaluation of response (through qualitative and quantitative assessment). Nevertheless, imaging assessment of response requires the use of standardized image acquisition, reconstruction techniques and reproducible and standardized quantitative response criteria.

#### 2.5.1. Radiological Imaging

The basic imaging for diagnosis, staging, therapy assessment and follow-up of NETs is computed tomography (CT). Contrast-enhanced CT (ceCT) is mandatory in patients and should be used in all subjects with adequate renal function and no other contraindications to iodinated contrast medium (CM), in order to highlight the characteristic enhancement of well-differentiated NETs. NETs are characterized by an earlier and more intense enhancement in the arterial phase compared to the surrounding normal parenchyma (especially pancreas and liver parenchyma) [78], due to their rich arterial blood network (hypervascularization). For this reason, a 3-phase examination of the liver is needed to search for hepatic metastases [79].

Several studies, focusing mainly on pancreatic NETs, have reported a correlation between contrast-enhancement patterns (CEP) and NET tumor grade in CT images. In particular, Cappelli et al. [80] studied the CEP of 60 pNETs resected in 52 consecutive patients and reported that the type of CEP was significantly related to tumor histology and tumor diameter (*p* < 0.001). In a cohort of 28 patients with pNET (13 G1 and 15 G2), Takumi et al. [81] showed that CT tumor conspicuity (absence of hyperattenuation during portal venous phase) was significantly associated with G2 pancreatic NETs (*p* = 0.016) with an accuracy of 71%. The accuracy of this pattern was subject to a bolus contrast administration technique and correct timing for arterial and venous phase acquisition. Furthermore, quantitative response assessment criteria depends on methodological issues, and a good practice is to measure lesions in the late arterial phase [79].

MR is preferred in young patients due to the lack of radiation exposure. Abdominal MR, including anatomical T2-weighted and T1-weighted pulse sequences with and without contrast and additional functional techniques (i.e., diffusion-weighted imaging and perfusion-weighted imaging), is established for diagnosis, staging, therapy assessment and follow-up of liver metastasis and primary pancreatic NETs. Moreover, bone and brain metastases are better assessed by MR than by CT [78]. As for CT, an also for MR, a multiphase contrast acquisition protocol is mandatory for NET characterization. In 2012, Manfredi et al. [82] reported that 60% of their entire cohort of 45 non-hyperfunctioning (NF) pNET showed hyperintensity on T2-weighted images and iso-/hypervascularity. Moreover, MR identified malignant NF-NETs with a sensitivity of 93.3% and a specificity of 76.9% (AUC = 0.85).

Ultrasonography (US) has an established role in imaging abdominal organs and in guiding biopsy. Its application therefore is mainly in the initial diagnosis and characterization of both liver metastases and pancreatic primaries and is therefore outside of the aim of this review.

As aforementioned, reproducible and standardized quantitative response criteria are necessary to evaluate therapy response through imaging biomarkers. Several criteria have been developed for this purpose since the introduction of the World Health Organization (WHO) criteria in 1979 [83]. The WHO criteria introduced the concept of measurable tumor burden at baseline CT, as the sum of the product of the largest perpendicular diameters of each detected lesion (bidimensional measurement), and of its percentage change, determined by two observations within at least 4 weeks. In 1992, the Southwest Oncology Group (SWOG), in cooperation with the National Cancer Institute, developed new response criteria, the SWOG criteria [66]. They differ from WHO criteria for: evaluation time window (3–6 weeks vs. 4 weeks); progressive disease definition (50% increase or an increase of 10 cm^2^ in the sum of products of all measurable lesions vs. a single lesion increased by >25%); complete response definition considered as normalization of specific tumor markers and other abnormal laboratory parameters not considered in WHO criteria.

In 2000, an international group of researchers developed the Response Evaluation Criteria in Solid Tumors (RECIST version 1.0) in order to simplify the evaluation of response to therapy and reduce error risks associated with the use of WHO and SWOG criteria [66]. In 2009, these criteria were modified (RECIST version 1.1) in order to improve their use and outcome in clinical trials and clinical practice [77]. RECIST 1.1 criteria are based on the evaluation of target lesions (up to 2 per organ and up to 5 in total) on CT or MR, through a one-dimensional measurement, the longest diameter of each target lesion (target lesion >1.5 cm in shortest diameter for lymph nodes and >1 cm in longest diameter for other lesions), which are summed up. In RECIST 1.1 criteria, partial response (PR) is defined as “at least a 30% decrease in the sum of diameters of target lesions, taking as reference the baseline sum diameters”. Progressive disease (PD) is defined as the “appearance of new lesions or at least a 20% increase in the sum of diameters of target lesions, taking as reference the smallest sum on study with an absolute increase of at least 5 mm”. This results in a 73% increase in volume, which is considerably higher than the volume increase needed to define PD by WHO or SWOG criteria. Table 3 summarized the radiological response criteria in solid tumor mentioned.

Although widely used, RECIST criteria have several limitations: some lesions, such as lymphangiosis or peritoneal carcinomatosis, or tumors with irregular margins are not reliably measurable; response assessment in cystic or osseous lesions is not reliable; the selection of target lesions could be arbitrary, particularly in multi-metastatic patients; response assessment is suboptimal for disease characterized by small lesions (<1 cm); therapy-induced necrosis may result in failure of target lesion shrinkage, mimicking stable or even progressive disease at RECIST evaluation [85].

All these limitations are particularly significant in NETs that are generally slow-growing tumors. Moreover, NETs eligible for PRRT are often characterized by the presence of several liver metastases, which could render the selection of target lesion difficult, and bone metastases and/or peritoneal metastases that cannot be measured reliably with RECIST criteria. Finally, an intrinsic effect of PRRT is the decreased intralesional vascularization and the subsequent development of intralesional necrosis, which may erroneously result in SD or even PD (in case of increasing size of target lesions or for the appearance of “new” lesions not detected before owing to their small size) by RECIST criteria.

Other criteria can probably better estimate the intralesional vascularization changes and differentiate between responding and non-responding lesions. These criteria should be adapted to the therapy response assessment of NETs. Further prospective trials are needed to validate different approaches.

The Choi criteria developed in 2007 for gastrointestinal stromal tumors (GIST) evaluation have been also proposed in NETs [84]. The Choi criteria use Hounsfield units (HU) for the evaluation of response, defining PR as a decrease in tumor attenuation by >15% or as a decrease in size (sum of diameters) by >10% and PD is defined as an increase in size by >10%. A recent retrospective study by Huizing et al. [86], including 44 patients with NETs (88.6% GEP, 6.8% lung and 2% of unknown origin) considered candidates for PRRT and aimed to assess the role of response evaluation methods in both anatomical (CT/MR) and functional ([^68^Ga]Ga-DOTA-TATE PET/CT) imaging and their prognostic capability for overall survival (OS). They analyzed images acquired prior to PRRT start, and images acquired 3 and 9 months after the last cycle, using RECIST 1.1, Choi and positron emission tomography response criteria in solid tumors (PERCIST). In a total of 110 lesions, stable disease was detected in 81.0% and 66.7% of patients with RECIST 1.1 criteria, and in 45.2% and 33.3% of patients with Choi criteria, at 3 and 9 months, respectively. They reported that the evaluation of PRRT response with Choi criteria lead to a longer mean OS in the responder group at 9 months compared to RECIST 1.1 analysis (42 versus 32 months), but similar results in the stable disease group (39 versus 37 months, respectively) and in the progressive disease group (27 versus 28 months, respectively). These findings indicate that Choi criteria may identify responders more accurately.

Change in arterial tumor attenuation and contrast enhancement is another criterion, which was recently evaluated by Pettersson et al. [87] in a retrospective cohort of 52 patients with metastatic pNETs who underwent ceCT at baseline, mid-treatment and at 3 months follow-up. Their study showed a significant decrease of the maximum arterial attenuation of liver metastases between baseline (217 ± 62 HU) and follow-up (198 ± 62 HU; *p* = 0.025). Moreover, the group suggested the use of a temporary intralesional increase in arterial tumor attenuation from baseline to mid-treatment as an early stage criterion of PRRT response. However, further studies are needed to validate this hypothesis.

To the best of our knowledge, only one retrospective study by van Vliet et al. [88] examined the abovementioned criteria with regard to progression-free survival (PFS) and overall survival (OS). In their cohort of 268 patients with gastroenteropancreatic and thoracic NENs treated with PRRT, response assessment by CT or MR imaging using RECIST 1.1 criteria, SWOG criteria, mRECIST criteria (where a minor response is defined by a decrease by 13–30%) and mSWOG criteria (where a minor response is defined by a decrease by 25–50%) were compared. Objective response (OR), stable disease (SD) and progressive disease (PD) were: 28%, 49% and 24%, respectively, for RECIST 1.1; 25%, 49% and 26%, respectively, for SWOG; 44%, 33% and 24%, respectively, for mRECIST; and 45%, 29% and 26%, respectively, for mSWOG. Despite these results, no significant differences were found in median PFS (26–30 months for OR, 27–34 months for SD and 8 months for PD) and median OS (55–57 months for OR, 56–74 months for SD and 11–12 months for PD) between the four response criteria. Hence, mRECIST and mSWOG criteria seem not to improve the accuracy of PRRT response assessment in NETs, while RECIST 1.1 and SWOG criteria yield comparable results. These data need prospective validation in a cohort studied also by functional imaging and, in this context, RECIST 1.1 appears to be most suitable for a comprehensive evaluation of PRRT response by morphological imaging [24,79].

#### 2.5.2. Functional Imaging

Functional imaging has a pivotal role for NET diagnosis, staging, therapy assessment and follow-up [79]. The radiotracers most widely used in NETs are listed in Table 4.

SSTR expression must be determined by functional whole-body imaging ([^68^Ga]Ga-DOTA-peptide PET/CT, if unavailable [^111^In]In-pentetreotide SPECT/CT) or immunohistochemistry in order to identify patients who are candidates for PRRT [12,98]. [^68^Ga]Ga-DOTA-peptide PET/CT is the mainstay for the assessment of somatostatin receptors (SSTR, subtypes 2a and 5) commonly expressed on the membranes of NET cells [99,100]. On the contrary, 2-[^18^F]fluoro-2-deoxy-D-glucose (2-[^18^F]FDG) PET/CT can serve as a surrogate for tumor metabolism. It is usually employed for the characterization of intermediate-grade and high-grade tumors [101]. For therapy assessment, functional imaging can highlight lesion properties that may be more reliable and predictive of PRRT outcome compared to data obtained by conventional imaging [79,102].

Several prognostic markers have been identified on baseline [^68^Ga]Ga-DOTA-peptide PET/CT in NETs patients prior to PRRT. Different papers highlighted the role of higher lesional uptake (SUVmax) as a marker predictive of response. In particular, Öksüz et al. [103] defined an SUVmax >17.9 on [^68^Ga]Ga-DOTA-TOC PET as a cut-off for favorable prognostic PRRT outcome in a cohort of 40 patients with advanced NETs treated with [^90^Y]Y-DOTATOC.

The Krenning score grades pathological uptake semiquantitatively: no uptake = score 0; very low = score 1; less than or equal to the liver = score 2; greater than the liver = score 3; greater than the spleen = score 4 [104]. A positive correlation exists between higher uptake (grade 3 or 4) and higher remission rate after therapy. Although it is applied also for [^68^Ga]Ga-DOTA-peptide PET/CT, the Krenning score was introduced for planar octreotide imaging, and for this reason other parameters were successively identified for PET/CT images, such as the tumor-to-spleen (T/S) ration and the tumor-to-liver (T/L) ratio. In a cohort of 30 NETs patients, Kratochwill et al. [105] evaluated three different prognostic parameters on [^68^Ga]Ga-DOTA-TOC PET: SUVmax >16.4 (sensitivity 95%, specificity 60.0%, AUC = 0.87), tumor-to-spleen (T/S) ratio >0.67 (sensitivity 95%, specificity 20%, AUC = 0.78) and tumor-to-liver (T/L) ratio >2.17 (sensitivity 95%, specificity 20%, AUC = 0.73) serve as cut-offs for favorable prognostic PRRT outcome. Moreover, an SUVmax >26.4 was identified as a good predictor for radiological response. On the other hand, in 2009 Gabriel et al. [106] did not find any benefit of an SUVmax analysis on [^68^Ga]Ga-DOTA-TOC PET for the prediction of individual therapy response.

In 2019, Sharma et al. [107] evaluated four [^68^Ga]Ga-DOTA-TATE PET/CT parameters in a cohort of 55 patients with metastatic NETs treated with [^177^Lu]Lu-PRRT: single lesion SUVmax, T/S and T/L ratios and SUVmax-av (defined as the average SUVmax of up to five target lesions in multiple organs sites) at baseline and follow-up PET/CT at the end of PRRT. Baseline single lesion SUVmax (SUVmax > 13.0) predicted both response (sensitivity 83%, specificity 84%, AUC = 0.78) and PFS (patients with SUVmax >13.0 had median PFS of 45.1 months compared to 19.9 months in patients with SUVmax <13.0). Baseline SUVmax-av (>10.2) was predictive of response (sensitivity 80%, specificity 83%, AUC = 0.78). Neither T/S nor T/L ratios were predictive of PRRT response.

Werner et al. [108] recently proposed SSTR-RADS, a structured reporting system for [^68^Ga]Ga-DOTA-peptide labeled-PET. The aim of this system is to standardize image assessment, both for diagnosis and for treatment guidance in NET patients, particularly with the purpose of patient selection for PRRT. According to their suggestion, PRRT should be considered in patients with an overall SSTR-RADS score of 4 (“positive uptake in site typical for NET lesions but lacking definitive finding on anatomic imaging”) or 5 (“intense uptake in site typical for NET with corresponding findings on conventional imaging”).

More recently, two new volumetric parameters were proposed by Abdulrezzak et al. [109]: the somatostatin receptor expressing tumor volume (SRETV) and the total lesion somatostatin receptor expression (TLSRE). Scarce pertinent literature reports a significant correlation between whole-body SRETV (SRETV_wb_) and disease progression after PRRT. Tirosh et al. [110] reported that “[^68^Ga]Ga-DOTA-TATE TV” (equalling SRETV_wb_) > 7.0 mL was significantly correlated with a higher risk for disease progression, while “[^68^Ga]Ga-DOTA-TATE TV” > 35.8 mL was associated with higher disease-specific mortality. Toriihara et al. [111] reported an association between “[^68^Ga]Ga-DOTA-TATE ∑SRETV” (corresponding to SRETV_wb_) > 11.29 mL and shorter progression-free survival.

Other prognostic markers were identified using a dual tracer approach consisting of baseline [^68^Ga]Ga-DOTA-SSTR PET/CT and baseline 2-[^18^F]FDG PET/CT in NETs patients prior to PRRT. In a cohort of 52 patients with NETs treated with [^177^Lu]Lu-PRRT, Severi et al. [112] reported that a negative baseline 2-[^18^F]FDG PET/CT scan is predictive of low tumor aggressiveness, while a positive baseline 2-[^18^F]FDG PET/CT scan with an arbitrary intralesional SUVmax cut-off of >2.5 in G2 grade NETs was associated with a more aggressive disease. The latter constellation may herald a greater benefit from systemic therapy other than PRRT. Similarly, an intralesional SUVmax >3 at baseline 2-[^18^F]FDG PET/CT was the only predictor of short progression-free survival (HR, 8.4; *p* < 0.001) in a larger prospective study by Binderup et al. on 98 patients with GEP and lung NETs [113]. The risk of death was higher in the 2-[^18^F]FDG PET-positive group. In a retrospective cohort of 60 patients with pNETs treated with [^177^Lu]Lu-PRRT, Sansovini et al. [114] reported a median PFS of 21.1 months for patients with a positive 2-[^18^F]FDG PET/CT, and a significantly (*p* < 0.0002) longer PFS of 68.7 months for patients with negative baseline 2-[^18^F]FDG PET/CT.

Considering the dual tracer approach, Chan et al. [115] developed the “NETPET score”, based on a qualitative evaluation of different intralesional and interlesional uptakes with the two radiotracers [^68^Ga]Ga-DOTA-SSTa and 2-[^18^F]FDG. The NETPET score correlates well with disease grade, overall survival and patient prognosis. NETPET score ranges from 1 ([^68^Ga]Ga-DOTA-SSTa+ and 2-[^18^F]FDG−) to 5 ([^68^Ga]Ga-DOTA-SSTa− and 2-[^18^F]FDG+), with score 1 indicating best prognosis and score 5 indicating worst prognosis. In their study, the median overall survival was not reached for subjects classified with NETPET score 1 or NETPET score 2–4, while for subjects classified with NETPET score 5 the median overall survival was 11 months (*p* = 0.0018). Furthermore, their study highlights the heterogeneity of NETs by the number of patients with G1 NETs but with a NETPET score greater than 1. Several studies have confirmed these data [116,117]. In particular, Thapa et al. [116] confirmed that high baseline 2-[^18^F]FDG SUVmax was associated with poor PRRT outcome in GEP-NETs, and also that 78% of responders showed high uptake on baseline [^68^Ga]Ga-DOTA-SSTa PET and low uptake or negative findings on baseline 2-[^18^F]FDG PET.

Several standardized quantitative criteria have been developed for therapy assessment with 2-[^18^F]FDG PET/CT imaging, as well as for morphological imaging. In 1999, the European Organization for Research and Treatment of Cancer (EORTC) criteria was the first PET scoring system created [118]. Later in 2009 these criteria were succeeded by more reproducible criteria, the positron emission tomography response criteria in solid tumors (PERCIST), modeled following RECIST criteria development [119]. Both EORTC and PERCIST criteria use the standardized uptake value (SUV) as the key parameter. The EORTC criteria uses the body weight SUV, while the PERCIST criteria uses SULpeak, defined as the average activity concentration within a 1 cm^3^ spherical volume of interest (VOI) centered on the “hottest focus” within the tumor, multiplied by the ratio of lean body mass (LBM) to injected activity decayed. Moreover, PERCIST criteria introduces the concept of metabolic tumor volume (MTV), which represents a measurement of tumor volume with increased glucose consumption, and total lesion glycolysis (TLG), that is the product of MTV and SUVmean. Table 5 summarized the EORTC and PERCIST criteria.

However, PERCIST use is not validated in [^68^Ga]Ga-DOTA-SSTa PET/CT imaging. Moreover, no reliable correlation between SUV change after PRRT and patient outcome has yet been proven in the aforementioned retrospective study of Huizing et al. [86], including 44 patients with NETs.

To the best of our knowledge, only one study evaluated the change (delta) of PET parameters in [^68^Ga]Ga-DOTA-TATE PET/CT. In 2010, Haugh et al. [120] evaluated SUVmax, T/S ratio and their delta between baseline and follow-up PET/CT 3 months after PRRT in a cohort of 33 patients. They showed that an early decrease in T/S ratio after the first cycle of treatment correlated with longer PFS, while delta SUVmax and delta CgA were no significant prognostic factors. However, these data need to be validated in future prospective studies with larger cohorts.

Finally, the role of 6-[^18^F]FDOPA PET/CT in PRRT response assessment needs to be mentioned. This radiotracer might be useful in patients with functioning GEP-NETs and negative somatostatin receptor PET due to the fact that 6-[^18^F]FDOPA uptake reflects cellular amino acid turnover rather than receptor density [92].

#### 2.5.3. Beyond RECIST, PERCIST and Conventional Imaging Parameters

In the few last years, the development of new technologies, acquisition protocols and methods of semiquantitative and quantitative image analysis (radiomics, deep learning and synthetic data) holds promise to support personalized clinical decisions, responding to the need to identify more standardized, reliable and specific parameters.

In particular, dynamic imaging is becoming of great interest. Adding dynamic acquisition to standard acquisition protocols has been proposed both in morphological and functional imaging. Dynamic contrast enhanced MR images (DCE-MRI), performed after injection of a contrast agent, provides information on microvessel perfusion, permeability and extracellular leakage space. Dynamic contrast-enhanced CT (DCE-CT) images provide information about blood flow, blood volume, capillary permeability and microvessel density. Dynamic functional imaging could allow for an absolute quantization of tracer uptake and kinetics over time. Finally, dynamic contrast-enhanced US (DCE-US) using microbubbles or nanoparticles to provide both morphologic and physiologic information at the same time [121].

Promising results have been reported for PRRT response assessment using DCE-MR. In 2012, Miyakazy et al. [122] evaluated DCE-MR imaging for monitoring and assessing treatment response in patients with neuroendocrine hepatic metastases (NET-HM) treated with [^90^Y]Y-PRRT. At baseline, the authors found lower whole liver distribution (*p* = 0.003) and higher tumor arterial flow fraction (*p* = 0.006) in responders compared to non-responders. In responders, tumor and whole-body distribution volume significantly increased after treatment, highlighting the limitations of PERCIST criteria. More recently, Weikert et al. [123] aimed to evaluate both diffusion-weighted imaging (DWI), including the apparent diffusion coefficient (ADC) and parameters related to intravoxel incoherent motion (IVIM), namely, diffusion coefficient (D), perfusion fraction (f) and pseudodiffusion coefficient (D∗), together with DCE-MR imaging in order to identify early treatment response in NETs with hepatic metastases treated with [^90^Y]Y-PRRT. For each PRRT cycle, MR images were performed before, 48 h after and 10 weeks after treatment. Moreover, an abdominal SPECT/CT was acquired 24 h after therapy. ANOVA analysis showed significant differences between responders and non-responders in mean ADC after 48 h (*p* = 0.026), with higher ADC and higher D at baseline in responding lesions (*p* = 0.023). Compared to baseline, a borderline significant (*p* = 0.046) increase in DCE-MR parameter v_e_ was also found in responding lesions after 10 weeks. In this study, SPECT/CT was performed after each PRRT cycle and a significant decrease in the L/S uptake ratio was found at second SPECT/CT scan in responding lesions compared to stable (*p* = 0.013) and progressive lesions (*p* = 0.021), where no such decrease was found.

Velikyan et al. [124] aimed to quantitatively and qualitatively compare the performance of dynamic [^68^Ga]Ga-DOTA-TOC and [^68^Ga]Ga-DOTA-TATE PET/CT in PRRT. Ten patients underwent one 45-min dynamic and three whole-body PET/CT examinations at 1, 2 and 3 h after injection with both tracers. Their study indicates that kinetic uptake parameters might reflect the receptor density more accurately than SUV-derived parameters: SUV parameters did not correlate with NET uptake rate (Ki) linearly, and an SUV > 25 corresponds to saturation for Ki (> 0.2 mL/cm^3^/min) and may not reflect the SSTR density accurately. These findings might partially explain the limited value delta SUV to monitor PRRT response. Moreover, the possibility to evaluate PRRT response with the use of Ki might reinforce the future use of dynamic PET/CT acquisition in this field.

Other innovative methods for quantification and image analysis derived from radiomics are expected to gradually translate into clinical medicine [125,126,127,128]. Using mathematical models for data characterization, radiomics allows us to extract a large number of features out of images, which might serve as prognostic parameters. These parameters might be applied in a clinical decision support system (CDSS) and guide therapeutic decisions in clinical research. In PRRT, the application of radiomics might bolster the use of PET, CT, MR and SPECT data as potential prognostic and theragnostic biomarkers. The impact of this resource might be fundamental, especially with regard to the use of targeted therapies and novel drugs. Several radiomic features have been shown to correlate with tumor heterogeneity and might harbor the potential to predict biological behavior and tumoral aggressiveness, eventually facilitating patient-tailored approaches [129,130,131]. Several studies have evaluated whether radiomic features (RFs) could be used for monitoring and assessing PRRT response in NETs patients.

In 2016, Wetz et al. [132] compared the Krenning score, T/L ratio and asphericity (ASP) between responding and non-responding lesions (total *n* = 66), segmented on baseline Octreoscan SPECT. In their study, a higher ASP level was associated with poorer response to PRRT. Moreover, ASP was the parameter with the highest AUC (>0.96) at 4 and 12 months of follow-up to discriminate responding from non-responding lesions, surpassing both the Krenning score and the T/L ratio.

In two subsequent studies, Werner et al. aimed to evaluate the prognostic value of radiomic features (RFs) extracted on baseline [^68^Ga]Ga-DOTA-SSTa PET/CT before PRRT treatment [133]. They reported that the textural feature entropy predicted both PFS and OS (cut-off = 6.7, AUC = 0.71, *p* = 0.02), while conventional PET parameters failed. This study had a comparably large and heterogenous cohort of 141 patients with NETs considered candidates for PPRT. In a further study from 2019 [134], they found a similar result in a smaller but more homogeneous cohort of patients (31 G1/G2 pancreatic NETs). At ROC analysis, entropy was predictive for OS (cutoff = 6.7, AUC = 0.71, *p* = 0.02), with increasing entropy predicting longer survival (entropy > 6.7, OS = 2.5 years, 17/31), while conventional PET parameters failed to predict patient outcome.

More recently, a study by Weber et al. [135] covering 304 lesions from 100 NET/NECs patients aimed to assess if conventional PET parameters and RFs derived from simultaneous [^68^Ga]Ga-DOTA-TOC PET/CT and MRI including ADC were associated with the proliferative activity of NETs, potentially allowing for a non-invasive tumor grading. In this study, conventional PET parameters and RFs of both [^68^Ga]Ga-DOTA-TOC PET/CT and ADC-MRI showed only a weak correlation with Ki-67. In 2020, the same group [136] aimed to assess changes in semiquantitative [^68^Ga]Ga-DOTA-TOC PET/CT and MRI parameters including ADC after different types of treatment including PRRT. In particular the authors evaluated if pre-therapeutic semiquantitative parameters extracted by [^68^Ga]Ga-DOTA-TOC PET/MRI could predict PRRT response in 9 patients. This study showed no value of any PET parameter for predicting PRRT response, despite the size being too small to harbor significance. However, responding patients showed a significant decrease in lesion volume on ADC maps and a borderline significant decrease in entropy.

Unfortunately, the results of these studies are still not transferable into clinical use. This is not only due to their methodology and in part to low patient numbers, but also to unresolved harmonization of all the pre-processing steps required to ensure repeatability and reproducibility of data and results. Several variables must be taken into account to achieve robust semiquantitative or radiomic features, such as variabilities in scanner hardware from different manufacturers, injected activity, acquisition time after injection for functional imaging, acquisition time per bed position, CT parameters used for attenuation correction of PET data, matrix size, slice thickness of reconstructed images, respiratory motion, PET reconstruction algorithm and other post-reconstruction steps, such as size of bin and segmentation methods used. This highlights the need for standardization, especially if such image analysis approaches are tested in multicenter studies [137,138,139,140].

### 2.6. Dosimetry

The impacts of the NETTER-1 study on NET treatment and the subsequent validation of the standard [^177^Lu]Lu-DOTA-TATE therapy protocol administering 7.4 GBq for 4 cycles every 8 weeks remain undiscussed. However, efforts of researchers to develop and validate individualized dosimetry protocols should get increasing attention in order to deliver personalized therapy. The objectives of a personalized dosimetry are:−the optimization of the absorbed dose to normal organs with an ad-hoc evaluation of toxicity for organs at risk (OAR), considering a maximum safe dose of 23 Gy to the kidney and 2 Gy to the bone marrow;−the optimization of the absorbed dose to tumors, developing strategies for interim imaging during therapy, for defining optimal time points for follow-up evaluation, and for planning the number of serial treatments based on post-therapy SPECT/CT owing to the partial decay of ^177^Lu into γ photons (energy of emission = 113 keV and 208 keV) [13,14,15,141].

In 2005, Pauwels et al. [142] identified for the first time an association between tumor shrinkage and dose estimate to tumors (Pearson R2 = 0.49) in a small cohort of 13 patients with GEP-NETs treated with [^90^Y]Y-PRRT. They reported a median absorbed dose of 232 Gy in responding tumors and 37 Gy in non-responding tumors. Their study identified an absorbed dose > 120 Gy as a cut-off to obtain a reduction in tumor size.

Another study by Ilan et al. [143] aimed to evaluate the dose–response relationship for metastatic pancreatic NETs (24 patients) treated with [^177^Lu]Lu-PRRT. They calculated the tumor-absorbed dose for 24 metastases (>2.2 cm in diameter), using sequential SPECT/CT imaging at 24, 96 and 168 h after PRRT infusion. A significant correlation between the absorbed dose and tumor shrinkage (according to RECIST 1.1) was observed. These results suggest that lesions receiving higher absorbed doses are more likely to respond to PRRT in terms of tumor size reduction. Nevertheless, NET metastases are very heterogeneous and in their study and some lesions with a higher absorbed dose did not follow this hypothesis, probably owing to contextual factors not accounted for (i.e., heterogeneity in binding affinity and receptor density, hypoxia, necrosis and proliferation rate). More recently, the same group [144] evaluated the dose-response relationship in metastatic small intestinal (SI) NETs (25 patients) treated with [^177^Lu]Lu-PRRT using the same approach. In contrast to their previous findings, all metrics regarding tumor shrinkage were unrelated to the absorbed dose, even if the administered activity correlated with tumor volume shrinkage (*p* = 0.01) and with RECIST 1.1 response (*p* = 0.01). Further studies in larger cohorts are necessary to analyze the dose-response association, stratified by different sites of NETs origin.

In 2017, Del Prete et al. [145] designed a personalized-PRRT (P-PRRT) protocol based on dosimetry in order to safely increase the absorbed dose to the tumor during a four-cycle induction course. The renal absorbed dose per injected radioactivity was predicted by the body surface area and glomerular filtration rate for the first cycle, and by renal dosimetry of the previous cycle for the following cycle. In this way, the prescribed renal absorbed dose of 23 Gy was reached across the four cycles. The authors estimated a lesion absorbed dose >130 Gy as a cut-off to obtain a reduction in tumor size. More important, the group found that the cumulative injected activity could safely be increased to 43.7 ± 16.5 GBq over four induction cycles with the P-PRRT protocol, increasing the tumor absorbed dose and potentially yielding therapeutic benefit without additional toxicity. These data corroborate previous findings and confirm that dosimetry is necessary to identify a personalized cumulative injected activity to limit toxicity for normal organs and to maximize tumor irradiation, even if further prospective studies are needed for validation and standardization.

Since personalized dosimetry using serial SPECT/CT is time consuming, the use of a pre-PRRT PET/CT scan with ^68^Ga-labeled peptides for dosimetry purposes has been evaluated. In 2012, Ezziddin et al. [146] compared, for the first time, standardized uptake values (SUVs) calculated on a pre-therapy [^68^Ga]Ga-DOTA-TOC PET/CT scan and absorbed doses of the subsequent first treatment cycle in 61 lesions in a consecutive cohort of 21 NENs (57.1% GEP and 42.9% lung) patients. The authors found a positive correlation between high lesional SUV (SUVmean > 15 and SUVmax > 25) and high tumor absorbed dose, suggesting that [^68^Ga]Ga-DOTA-TOC PET/CT imaging may predict PRRT response and help select patients. However, these data need validation through further prospective studies.

## 3. Conclusions

The present review outlines the importance and the limitations of different biomarkers, summarized in Table 6, in a multidisciplinary approach in order to improve both patient selection for PRRT and evaluation of PRRT response in patients with neuroendocrine tumors.

Morphological and functional imaging will assume an important role as biomarker source, particularly with the advent of innovative image analysis methods. However, there is still a need to identify and validate reliable parameters for this purpose. The combination of imaging biomarkers with circulating biomarkers and new genomic multianalyte biomarkers may lead the way for future individualized strategies for PRRT patients.

## Figures and Tables

**Figure 1 diagnostics-10-01083-f001:**
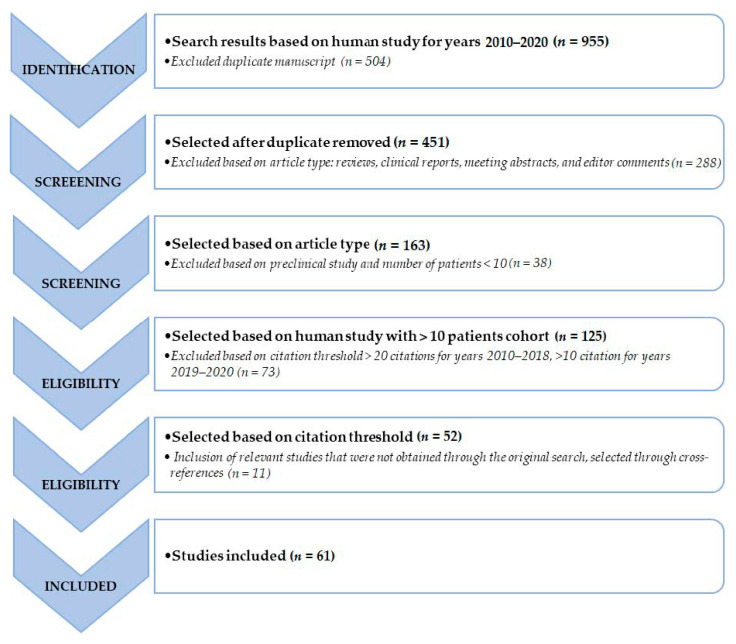
Schematic representation of the performed literature search and the review strategy.

**Table 1 diagnostics-10-01083-t001:** Classification and grading criteria for GEP-NENs, according to the WHO 2019 Grading Classification.

Classification	Differentiation	Nomenclature	Grade	Mitotic Rate (mitoses/2 mm^2^)	Ki-67 Index
NETs	Well differentiated	NET, G1	Low	<2	<3%
		NET, G2	Intermediate	2–20	3–20%
		NET, G3	High	>20	>20%
NECs	Poorly differentiated	NEC, small-cell type (SCNEC)	High	>20	>20%
		NEC, large-cell type (LCNEC)	High	>20	>20%
MiNEN	MiNEN	Well or poorly differentiated	Variable	Variable	Variable
− *SCNEC*: small-cell neuroendocrine carcinoma; *LCNEC*: large-cell neuroendocrine carcinoma; *MiNEN*: mixed neuroendocrine-non-neuroendrocrine neoplasm; − mitotic rates are determined by counting in 50 fields of 0.2 mm^2^; − the *Ki-67* proliferation index is determined by counting at least 500 cells in the regions of highest labelling (hot-spots), which are identified at scanning magnification.					

**Table 2 diagnostics-10-01083-t002:** Characteristics of theragnostic radionuclides used in patients with NENs.

Radionuclide	Half-Life	Type of Emission	Energy of Emission (keV)	Particle Range (R) in Soft Tissue
^90^Y (yttrium)	2.67 d	β-	934	Rmax: 11.3 mm
^177^Lu (lutetium)	6.65 d	β- γ γ	134 113 208	Rmax: 2 mm Rmean: 0.5 mm
^161^Tb (terbium)	6.89 d	β- γ γ	154 49 75	Rmax: 0.29 mm
^225^Ac (actinium)	10.0 d	α α α α α γ	5.637 5.732 5.791 5.793 5.830 99.8	Rmax: 45–85 μm
^213^Bi (bismuth)	45.6 min	α α γ	5.558 5.875 324	Rmax: 45–85 μm
[18,19,20]				

**Table 3 diagnostics-10-01083-t003:** Main radiological response criteria in solid tumor.

Response Criteria	WHO [83]	SWOG [66]	RECIST 1.1 [77]	CHOI [84]
Complete response (CR)	Disappearance of all known lesions. Determined by two observations not less than 4 weeks apart	Disappearance of all measurable and evaluable lesions without any new lesions or disease-related symptoms. Determined by two observations not less than 3–6 weeks apart	Disappearance of all target and non-target lesions, without any new lesions. Any pathological lymph nodes must have reduction in short axis to <10 mm. Determined by two observations not less than 4 weeks apart.	Disappearance of all lesions, without any new lesions.
Partial response (PR)	Sum of product of all lesions decreased by >50% for at least 4 weeks; no new lesions; no progression of any lesions.	Sum of product of all lesions decreased by >50% for at least 3–6 weeks; no new lesions; no progression of evaluable lesions.	At least 30% decrease of the sum of maximum diameters of target lesions; no new lesions; no progression of disease.	Decrease in size of >10% or decrease in tumor density (HU) >15% on CT; without any new lesions. No obvious progression of non-measurable disease.
Stable disease (SD)	Sum of product of all lesions decreased by <50% or increased by <25% in the size of one or more lesions.	Sum of product of all lesions decreased by <50% or increased by <50% or 10 cm^2^ for at least 3–6 weeks.	Does not meet the criteria for CR, PR or PD, taking as reference the smallest sum of maximum diameters of target lesions.	Does not meet the criteria for CR, PR or PD. No symptomatic deterioration attributed to tumor progression.
Progressive disease (PD)	A single lesion increased by >25% (over the smallest measurement achieved for the single lesion) or the appearance of new lesions.	50% increase or an increase of 10 cm^2^ in the sum of products of all measurable lesions over the smallest sum observed; clear worsening of any evaluable disease; appearance of new lesion.	Sum of the maximum diameter of lesions increased by >20% over the smallest achieved sum of maximum diameter. The appearance of one or more new lesions is always considered progression.	Increase in tumor size of >10% and does not meet criteria of PR by tumor density (HU) on CT. The appearance of one or more new lesions is always considered progression.

**Table 4 diagnostics-10-01083-t004:** Characteristics of the clinically most widely used radiotracers for functional imaging of NENs in adult patients.

	Half-Life	Injection Activity	Acquisition Time after Injection	Mean Effective Dose Equivalent (mSv/MBq)	Physiological Biodistribution	Excretion	SSTR Affinity Profile
[^111^In]In-DTPA-octreotide	67.8 h	185 to 222 MBq	4, 24, 48 h	0.054	High uptake in the spleen, kidney, liver, bowel and gallbladder; faint uptakey in adrenal glands, pituitary and thyroid glands.	Clearance with 50% and 85% of the injected dose through urinary excretion by 6 and 24 h, respectively. Hepatobiliary excretion (2%) and spleen trapping (2.5%).	SSTR1 > 10,000; SSTR2 22 ± 3.6; SSTR3 182 ± 13; SSTR4 >1000; SSTR5 237 ± 52
[^68^Ga]Ga-DOTA-TATE	68.3 min	100 to 200 MBq	45–60 min	0.0257	Intense accumulation in the spleen, kidneys, and adrenal, salivary, and pituitary glands. Accumulation in liver is less intense than in spleen. Thyroid is faintly visible. Variable tracer uptake in uncinate process of pancreas.	Clearance with 40% and 75% of the injected dose through urinary excretion by 3 and 24 h, respectively. Less than 2% of the injected dose is excreted in the faeces by 48 h after injection.	SSTR1 > 10,000; SSTR2 0.2 ± 0.04; SSTR3 > 1000; SSTR4 300 ± 140; SSTR5 377 ± 18
[^68^Ga]Ga-DOTA-TOC	68.3 min	100 to 200 MBq	60–90 min	0.023			SSTR1 > 10,000; SSTR2 2.5 ± 0.5; SSTR3 613 ± 140; SSTR4 >1000; SSTR5 73 ± 12
[^68^Ga]Ga-DOTA-NOC	68.3 min	100 to 200 MBq	60–90 min	0.025			SSTR1 > 10,000; SSTR2 1.9 ± 0.4; SSTR3 40 ± 5.8; SSTR4 260 ± 74; SSTR5 7.2 ± 1.6
2-[^18^F]FDG	109.8 min	From 14 to 7 (MBq·min·bed − 1·kg − 1) × patient weight (kg)/emission acquisition duration per bed position (min·bed − 1).	45–50 min	0.019	Cerebral gray matter, salivary glands, lymphatic tissue including Waldeyer’s ring, muscles, brown adipose tissue, myocardium, liver, kidneys, collecting system and bladder, gastrointestinal tract, testes, and ovaries show physiological 2-[^18^F]FDG uptake.	Majority is excreted unaltered by the kidneys; 20% of the injected dose is recovered in the urine within 2 h.	Not applicable

Literature references [7,11,89,90,91,92,93,94,95,96,97].

**Table 5 diagnostics-10-01083-t005:** Main PET scoring system for functional response criteria in solid tumor.

Response Criteria	EORTC [118]	PERCIST [119]
Complete metabolic response (CMR)	Complete resolution of 2-[^18^F]FDG uptake within all lesions, making them indistinguishable from surrounding tissue.	Complete resolution of 2-[^18^F]FDG uptake within all lesions, to a level of less than or equal to that of the mean liver activity and indistinguishable from the background (blood pool uptake).
Partial metabolic response (PMR)	Reduction of at least 25% in the sum of SUV uptake of all lesions detected at baseline.	Reduction of at least 30% in the sum of SULpeak of all target lesions detected at baseline and an absolute drop of 0.8 SULpeak units.
Stable metabolic disease (SMD)	Does not meet the criteria for CR, PR or PD.	Does not meet the criteria for CR, PR or PD.
Progressive metabolic disease (PMD)	Increase of at least 25% in the sum of SUV uptake of all lesions detected at baseline. The appearance of one or more new FDG-avid lesions that are typical for cancer and not related to inflammation or infection is always considered progression.	Increase of at least 30% in the sum of SULpeak of all target lesions detected at baseline and an absolute increase of 0.8 SULpeak units. Or 75% increase in total lesions glycolysis (TLG), with no decrease in SUL. Or The appearance of one or more new FDG-avid lesions that are typical of cancer and not related to inflammation or infection is always considered progression.
Lesion measurability	Standard uptake value (SUV) of lesion with high 2-[^18^F]FDG uptake.	SULpeak at baseline lesions at least 1.5 higher than liver SULmean (+2DS) or 2.0 higher than blood pool SULmean.

**Table 6 diagnostics-10-01083-t006:** Definition of a biomarker and overview of PRRT response biomarkers in GEP-NETs.

Biomarker [26,27,28]	Definition	
Diagnostic	To help to diagnose/detect cancer, as in the case of identifying early stage cancers	
Prognostic	Provide the aggressiveness of a pathology, as in the case of determining the patient’s ability to survive without treatment	
Predictive	Predict how well a patient will respond to treatment	
Clinical Biomarker	Background	**Considerations**
HRQoL scale	HRQoL is an evaluation of QOL and its relationship with health; includes not only wealth and employment but also the built environment, physical and mental health, education, recreation and leisure time and social belonging.	Several studies with varying assays and populations. Moderate metrics, even if not specific.
EORTC QLQ-C30 and GI-NET21	EORTC QLQ-C30 and GI-NET21 are quality-of-life questionnaires specific for NET, evaluating gastrointestinal symptoms, any factors related to cancer, psychosocial problems, treatment side effects and other events.	Several studies with varying assays and populations. Moderate metrics.
Pathological Tissue Biomarkers	Background	**Considerations**
Ki-67	Ki-67 plays a more prominent role in NETs compared to other tumors, because of the wide disparity in biological behavior between different grades of disease.	Essential to define tumor grade and to enroll patients for PRRT. Low metrics for controversially results as a prognostic biomarker for PRRT response.
Circulating Biomarkers	Background	**Considerations**
CgA	Chromogranin A is acidic glycoprotein of 439 amino acids, released from neurons, neuroendocrine cells and NET cells. Elevated CgA levels represent a surrogate marker for tumor burden.	Elevated CgA levels > 600 ng/mL were associated with earlier tumor progression in a few studies. CgA prognostic role on PRRT response assessment has not yet been evaluated prospectively. Low/moderate metrics as a prognostic biomarker for PRRT response.
NSE	NSE is a glycolytic enzyme typically present in the cytoplasm of neurons and neuroendocrine cells.	NSE > 15 ng/mL independently predicted shorter overall survival in metastatic GEP-NET after PRRT in one study. Poor metrics as a prognostic biomarker for PRRT response for limited studies.
5-HIAA	5-HIAA represents the main metabolite of the amine derivate serotonin, overproduced in case of carcinoid syndrome, mainly by small intestinal NETs with liver metastases. Patients with carcinoid symptoms related to serotonin hypersecretion have shorter PFS after PRRT.	Essential to monitor patients with carcinoid syndrome. Possible moderate metrics as a prognostic biomarker for PRRT response but limited studies. Further prospectively trials are needed.
IBI	IBI is a new inflammation circulating biomarker, which is derived from serum C-reactive protein (CRP) and albumin levels.	Two studies reported that normal baseline IBI predict better outcome in NET patients treated with PRRT. Possible moderate metrics as a prognostic biomarker for PRRT response but limited studies.
Genomic Multianalyte Biomarkers	Background	**Considerations**
NETest	NETest is a multi-transcript molecular signature of 51 specific NET genes for PCR-based blood analysis.	Prospective studies with excellent metrics. Limited availability and comparably high costs.
PPQ test	PPQ is an algorithm that integrates blood-derived NETspecific gene transcripts with tissue Ki-67 values.	Prospective studies with excellent metrics. Limited availability and comparably high costs. The PPQ is not yet validated as a prognostic marker of survival.
Imaging Biomarkers	Background	**Considerations**
CeCT	Anatomical imaging using X-ray and contrast enhancement.	Several studies with varying assays and populations. High/moderate metrics due to the use of RECIST criteria, even if with several limitations.
MR	Anatomical imaging using MR, typically with a multiphase contrast acquisition protocol.	Essential for the evaluation of liver metastases. Promising moderate metrics as a prognostic biomarker for PRRT response for dynamic contrast-enhanced and radiomic information, but limited studies. Further prospectively trials are needed.
[^68^Ga]Ga-DOTA-SSTa	PET imaging of somatostatin receptor by positron-emitting radionuclide	Current gold standard for functional imaging in GEP-NETs. Extensively studied and with excellent metrics. High metrics due to the assessment of somatostatin receptors, in terms of SUV uptake and tumor burden.
2-[^18^F]FDG	PET imaging of glucose metabolism by positron-emitting radionuclide	Negative baseline 2-[^18^F]FDG PET/CT scan is predictive for low tumor aggressiveness, while a positive baseline 2-[^18^F]FDG PET/CT scan (SUVmax > 2.5) in G2 NETs is associated with a more aggressive disease. Possible moderate metrics as a prognostic biomarker for PRRT response but limited studies. Further prospectively trials are needed.
Dosimetry	Background	**Considerations**
[^177^Lu]Lu-SPECT imaging	SPECT imaging of somatostatin receptor by γ- emitting of [^177^Lu]Lu	Possible moderate metrics as a predictive biomarker for PRRT response but limited studies. Further prospectively trials are needed.

**NOTE**: **5-HIAA** = 5-hydroxyindole acetic acid; **CgA** = chromogranin A; **CeCT** = contrast-enhanced computed tomography; **EORTC QLQ-C30 =** European Organization for Research and Treatment of Cancer quality-of-life questionnaires C-30; **HRQoL** = health-related quality of life; **IBI =** inflammation-based index; **MR** = magnetic resonance; **NSE** = neuron-specific enolase; **PPQ** = PRRT predictive quotient; **SSTa** = somatostatin analogues.

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
