# Peer review of "The Challenge of Evaluating Response to Peptide Receptor Radionuclide Therapy in Gastroenteropancreatic Neuroendocrine Tumors: The Present and the Future"

_diagnostics, 2020, doi:10.3390/diagnostics10121083_

Round 1

Reviewer 1 Report

I very much enjoyed reading this excelent review article on this very topical area.

Overall well written and comprehensive on all aspects of relevant biomarkers in regard to the assessment of PRRT response.

A few comments and suggestions:

1. There is a general assumption that the readers need to be made aware the definition of what a biomarker is and the definitions of the various imaging response criteria: Tables to defines these would be clearer 

1 Methods: An additional exclusion was based on citation threshold > 20 citations for years 2010-2018 and > 10 citation for years 2019-2020: Please explain this exclusion 

2. Pathological tissue biomarkers: A separate subsection of site of primary would be of use 

3. 2.4.1  Serum biomarkers: Is Pancreatic polypeptide or bradykinin associated with a specific hormone syndrome?? Is there a suggestion that all patients must be screened for possible functional syndromes??

4. NETest: In regard to PRRT response is heavily dependant on additional mathematical manipulations to achieve the noted correlations - in this regard how useful it is - it also a commercial test- limited to a few. Prospective trials are required to validate its true role in this setting - overall the conclusions here are too strong.

5. There is an overall consistent lack of critical analysis of each of the biomarkers in each category- re their relative utility and value and their future role. This needs to be strengthened to add weight to this paper. Summation in this regard with Table would be of great value. 

Author Response

Reviewer #1:

1) There is a general assumption that the readers need to be made aware the definition of what a biomarker is and the definitions of the various imaging response criteria: Tables to defines these would be clearer

A: We thank the reviewer for the comment. We have added a final table (Table 6) to summarize all the biomarkers according to the text and defining if diagnostic, prognostic or predictive and two tables (Table 3 and Table 5) to summarize morphological and PET imaging response criteria.

2) Methods: An additional exclusion was based on citation threshold > 20 citations for years 2010-2018 and > 10 citation for years 2019-2020: Please explain this exclusion

A: We thank the reviewer for the comment. We have added an explanation for this criterion in the text; due to the huge amount of manuscripts identified through our research on the subject, we have considered adding this criterion in order to select the most significant and impactful papers on PRRT response assessment.

3) Pathological tissue biomarkers: A separate subsection of site of primary would be of use.

A: We thank the reviewer for the comment. According to reviewer comment we divided the Pathological tissue biomarkers paragraph in two separate sections, namely Ki67 and grading (session 1) and primary origin (session 2), respectively.

4) Serum biomarkers: Is Pancreatic polypeptide or bradykinin associated with a specific hormone syndrome?? Is there a suggestion that all patients must be screened for possible functional syndromes??

A: We thank the reviewer for the comment.  The evidence concerning Pancreatic polypeptide and/or bradykinin are low.  Concerning functional syndrome, we agree with the reviewer that all patients should be screened for specific biomarkers. Nevertheless, they could be not available in all centers. We added a comment on this point in the Serum biomarkers section.

5)  NETest: In regard to PRRT response is heavily dependant on additional mathematical manipulations to achieve the noted correlations - in this regard how useful it is - it also a commercial test- limited to a few. Prospective trials are required to validate its true role in this setting - overall the conclusions here are too strong.

A: We agree with the reviewer on this comment.  We added a paragraph on the limits of NETest and made the conclusion less strong.

6) There is an overall consistent lack of critical analysis of each of the biomarkers in each category- re their relative utility and value and their future role. This needs to be strengthened to add weight to this paper. Summation in this regard with Table would be of great value.

A: We thanks the reviewer for the comment. We have added a table (Table 6) to summarize all the biomarkers cited in the text, their application and limits.

Reviewer 2 Report

This manuscript was conducted to outline the challenge of evaluating response to PRRT in GEP-NEN. The authors went into detail on biomarkers, morphological-/ functional imaging and dosimetry and provided a comprehensive review of present parameters for this purpose. It is well written and will be of interest to the NET community. Nonetheless there are some aspects that need further clarification and discussion.

Comments:

  • Radiotracer need a proper definition in the first place, after that an easy abbreviation can be used, e.g. [68Ga]Ga-DOTATATE. Please find the ''Tracer Nomenclature'' (https://www.eanm.org/content-eanm/uploads/2019/12/EANM_GUIDANCE-_TRACER_NOMENCLATURE-1.pdf)
  • Line 47 ff. The increasing incidence of NENs is not only owing to remarkable progress in functional imaging but also to further increasing awareness of the occurrence of NEN.
  • Line 207 ff. The authors discuss intertumoral and intratumoral heterogeneity of Ki-67 expression, which can be an explanation for fluctuation in some patients. Unfortunately, the Ki-67 delivers more limitations/ pitfalls. Assessment of Ki-67 depends on expertise of the reporting pathologist and core biopsies represent only a small tumor sample, which impedes accurate heterogeneity assessment, especially of intermediate G2-lesion.
  • Line 253: Reference 50 is missing in the context.
  • Line 316 (RECIST criteria) Reference missing
  • Line 344 (RECIST 1.1 criteria) Reference missing
  • Line 499: Incorrect reference Krenning Score - Kwekkeboom DJ et al J Clin Oncol (2005) vs. Wetz et al.
  • Reference 108 (Bozkurt et al. Eur. J. Nucl. Nucl. Med. Mol. Imaging 2017) is missing. Even if the role of 18F-DOPA in the prediction of individual therapy has yet not been proven, in a pre-/ or post-therapeutic setting 18F-DOPA may be helpful in the identification of weakly or no lesional SSTR-expression, thus hinting at an aggravating outcome in affected patients. This could be worse mentioning.
  • Line 640: … with poorer response to PRRT?. Why a question mark?
  • Line 660: Reference 123: 68Ga-DOTATOC-PET/MRI - 29 patients with pairs of 68Ga-DOTATOC-PET/MRI scans, however only 9 cases were treated with PRRT). The sample size is way too small and did not meet the inclusion criteria (studies > 10 Patients).

Author Response

Reviewer #2:

1) Radiotracer need a proper definition in the first place, after that an easy abbreviation can be used, e.g. [Ga]Ga-DOTATATE. Please find the ''Tracer Nomenclature'' (https://www.eanm.org/content-eanm/uploads/2019/12/EANM_GUIDANCE-_TRACER_NOMENCLATURE-1.pdf)

A: As correctly pointed out by the reviewer, we checked radiotracer definition throughout all the manuscript according to EANM guidance.

2) Line 47 ff. The increasing incidence of NENs is not only owing to remarkable progress in functional imaging but also to further increasing awareness of the occurrence of NEN

A: We agree with the reviewer comment and we have changed the sentence according to the suggestion.

3) Line 207 ff. The authors discuss intertumoral and intratumoral heterogeneity of Ki-67 expression, which can be an explanation for fluctuation in some patients. Unfortunately, the Ki-67 delivers more limitations/pitfalls. Assessment of Ki-67 depends on expertise of the reporting pathologist and core biopsies represent only a small tumor sample, which impedes accurate heterogeneity assessment, especially of intermediate G2-lesion.

A: We thank the reviewer for his/her comment. As correctly pointed out by the reviewer, we add a sentence on Ki67 limits.

4) Several references error. Namely,

- Line 253: Reference 50 is missing in the context.

- Line 316 (RECIST criteria) Reference missing

- Line 344 (RECIST 1.1 criteria) Reference missing

- Line 499: Incorrect reference Krenning Score - Kwekkeboom DJ etal J Clin Oncol (2005) vs. Wetz et al.

A: we thank the reviewer for pointing out these errors, we have revised all the references through the text.

5) Reference 108 (Bozkurt et al. Eur. J. Nucl. Nucl. Med. Mol. Imaging2017) is missing. Even if the role of F-DOPA in the prediction of individual therapy has yet not been proven, in a pre-/ or post-therapeutic setting F-DOPA may be helpful in the identification of weakly or no lesional SSTR-expression, thus hinting at an aggravating outcome in affected patients. This could be worse mentioning.

A: We thank the reviewer for this comment. As correctly pointed out by the reviewer, we added a sentence also on 18F-DOPA PET with this intent.

9) Line 640: … with poorer response to PRRT? Why a question mark?

A: We thank the reviewer for this comment. It was a typing error, we removed it.

10) Line 660: Reference 123: Ga-DOTATOC-PET/MRI - 29 patients with pairs of Ga-DOTATOC-PET/MRI scans, however only 9cases were treated with PRRT). The sample size is way too small and did not meet the inclusion criteria (studies > 10 Patients).

A: We thank the reviewer for this comment. We agree that this study did not meet all inclusion criteria and we underlined this aspect in the text. However, we cited this paper because in the radiomics field there are still few studies. We believe that these few articles may all be of interest to readers with particular regard to the methodological aspect, which tends to be the weak point of radiomics studies.
